# Mdm2-Mediated Downmodulation of GRK2 Restricts Centrosome Separation for Proper Chromosome Congression

**DOI:** 10.3390/cells10040729

**Published:** 2021-03-25

**Authors:** Clara Reglero, Belén Ortiz del Castillo, Verónica Rivas, Federico Mayor, Petronila Penela

**Affiliations:** 1Departamento de Biología Molecular and Centro de Biología Molecular “Severo Ochoa” (UAM-CSIC), 28049 Madrid, Spain; cr3023@cumc.columbia.edu (C.R.); belen.ortiz@cbm.csic.es (B.O.d.C.); vrivas@cbme.csic.es (V.R.); fmayor@cbm.csic.es (F.M.J.); 2Instituto de Investigación Sanitaria La Princesa, 28006 Madrid, Spain; 3CIBER de Enfermedades Cardiovasculares, ISCIII (CIBERCV), 28029 Madrid, Spain

**Keywords:** centrosome separation, mitotic spindle, Mdm2, GRK2, MST2, protein degradation, ubiquitination, phosphorylation

## Abstract

The timing of centrosome separation and the distance moved apart influence the formation of the bipolar spindle, affecting chromosome stability. Epidermal growth factor receptor (EGFR) signaling induces early centrosome separation through downstream G protein-coupled receptor kinase GRK2, which phosphorylates the Hippo pathway component MST2 (Mammalian STE20-like protein kinase 2), in turn allowing NIMA kinase Nek2A activation for centrosomal linker disassembly. However, the mechanisms that counterbalance centrosome disjunction and separation remain poorly understood. We unveil that timely degradation of GRK2 by the E3 ligase Mdm2 limits centrosome separation in the G2. Both knockout expression and catalytic inhibition of Mdm2 result in GRK2 accumulation and enhanced centrosome separation before mitosis onset. Phosphorylation of GRK2 on residue S670 enables a complex pattern of non-K48-linked polyubiquitin chains assembled by Mdm2, which correlate with kinase protein degradation. Remarkably, GRK2-S670A protein fails to phosphorylate MST2 despite overcoming Mdm2-dependent degradation, which results in defective centrosome separation, shorter spindles, and abnormal chromosome congression. Conversely, extra levels of wild-type kinase in the G2 cause increased inter-centrosome distances with longer spindles, also converging in congression issues. Our findings show that the signals enabling activity of the GRK2/MST2/Nek2A axis for separation also switches on Mdm2 degradation of GRK2 to ensure accurate centrosome dynamics and proper mitotic spindle functionality.

## 1. Introduction

Abnormal numbers, attachment or movement of chromosomes are all causes of defective chromosome distribution, eventually leading to aneuploidy and cancer [1,2]. The spindle apparatus’s bipolarity, along with an adequate inter-pole distance and orientation, is essential for the accurate segregation of chromosomes [3]. It is now well appreciated that the centrosome, the main microtubule (MT)-organizing center (MTOC) in most animal cells consisting of two centrioles surrounded by a pericentriolar protein matrix, plays a major role in allowing correct spindle positioning and organization [4,5].

Two different types of connections between centrioles regulate centrosome duplication and separation, which are events coordinated with the cell cycle that occurs once in every cell cycle as part of the centrosome cycle [6]. The S-M linker prevents centriole reduplication, and its removal at mitosis/G1 transition (“centrosome disengagement”) licenses the centrosome for duplication in the next S phase of the cell cycle [7,8]. The second connection, the so-called G1-G2 tether, is a complex proteinaceous structure composed of cNap1 and rootletin proteins, among others, which all together keep the parental centrioles joined from G1 to late G2 [9,10]. During G2, this linker disassembles (“centrosome disjunction”) in a process governed by cell cycle kinases CDK1, aurora A, PLK1 and NIMA-related kinase Nek2, being this latter responsible for phosphorylation of C-Nap1 and rootletin to allow centrosomes separation and subsequent movement in a kinesin (Eg5)-mediated manner, allowing the formation of the two spindle poles [9,11].

Centrosome separation may proceed with different timing and protein requirements through the prophase of the prometaphase pathways. The first is completed before nuclear envelope breakdown (NEBD) involving the interaction between MTs and the nuclear envelope, whereas the second pathway taking place in prometaphase requires myosin activity at the cell cortex [9,12]. Improper timing of centrosome disjunction and separation through these pathways may result in monopolar or bipolar mitotic spindles with geometry defects in metaphase [12,13,14]. The ubiquitin–proteasome system controls many aspects of the centrosome cycle beyond its general role in protein recycling. Both the proteasome itself and SCF ubiquitin ligase complexes localize to the centrosome, and different examples of their relevance for control of centriole duplication have been reported (reviewed in [15,16]). Except for the well-characterized destabilization of Nek2A and kinesin Eg5 proteins by cell cycle ligase complexes APC/Cdc20 and APC/Cdh1, respectively [17,18], ubiquitin regulation of centrosome separation is less explored, and the involvement of other E3 ligase activities largely unknown. Studies have been focused instead on complex kinase cascades that regulate effector proteins for the split and separation of centrosomes. Thus, the activity of Nek2A is strictly antagonized by phosphatases PP1α and PP1γ or pericentrin [19,20], while promoted by MST2, a component of the Hippo pathway that directly phosphorylates and activates Nek2A [21]. Growth factors increase the early separation of the centrosome in the prophase pathway by activating the kinase module MST2-Nek2 [22]. Recently, G protein-coupled receptor kinase GRK2 has been found to be specifically involved in EGF-induced centrosome separation by means of the phosphorylation of MST2 and activation of Nek2A in a PLK1-independent manner [23].

GRK2 is a multifunctional protein that modulates signaling mediated by G protein-coupled receptors (GPCRs) and via phosphorylation or scaffolding interactions with an increasing number of non-GPCR substrates partners, which are engaged in key cellular processes [24,25]. GRK2 its tightly regulated and altered GRK2 levels and activity are reported in pathological contexts, such as metabolic dysfunctions and several types of cancer [26,27]. GRK2 functionality is mainly controlled by posttranslational protein modifications that affect catalytic activity, drive the kinase towards different substrates or modulate its subcellular location or proteolysis rate (reviewed in [28,29,30]. We have reported that GRK2 levels gradually decline during the G2 by default, but in genotoxic conditions, the protein is instead stabilized as part of the G2 checkpoint contributing to delay the cell cycle progression [31]. GRK2 degradation in the G2 involves phosphorylation by CDK2–cyclin A and the subsequent binding of the prolyl-isomerase Pin1, a critical cell cycle regulator [31,32]. Both abundance and activation status of GRK2 are known to affect EGFR–mediated separation of duplicated centrosomes in the G2 [23], but whether the sophisticated control of GRK2 stability plays a role has not been addressed. Our group and others have identified Mdm2 as the main E3 ligase implicated in GRK2 turnover upon GPCR stimulation [33,34,35], although the E3 ligase targeting GRK2 during the G2 remains to be identified [31]. Interestingly, downregulation of the centrosome-associated protein CEP131 by Mdm2 has been associated with centrosome amplification [36,37]; however, no effects in centrosome separation have been described so far. Our results show that control of GRK2 stability by Mdm2 is an intrinsic event in the regulation of centrosome separation that counterbalances EGF-induced and GRK2-mediated activation of the MST2–Nek2A module. We also observed that the signal-enabling activity of the GRK2/MST2/Nek2A axis for separation switch on Mdm2’s polyubiquitination of GRK2. Moreover, we demonstrate that interference with GRK2 degradation results in abnormal centrosome separation, and accordingly triggering defects in mitotic spindle length and chromosome congression.

## 2. Material and Methods

### 2.1. Cell Culture, Transfection and Treatments

HeLa and HEK293-T cells were obtained from the American Type Culture Collection (ATCC). Wild-type and Mdm2/p53 double-null mouse embryo fibroblasts (MEF) (MEF Mdm2-KO) cells were gently provided by Dr. G. Lozano (Anderson Cancer Center, Houston). Cells were cultured in Dulbecco’s modified Eagle’s medium (DMEM) supplemented with 10% (*v/v*) fetal bovine serum (FBS) at 37 °C in a humidified 5% CO2 atmosphere. HeLa cell lines stably overexpressing wild-type GRK2 (HeLa-WT5) or mutant construct S670A (HeLa-A1) were previously described [31]. MEF-Mdm2 knock-in (KI) cells were generated by stable expression of FLAG-tagged human Mdm2 in Mdm2-KO MEF cells. The pCMV–FLAG–Mdm2 construct kindly provided by Dr. L. Mayo (Indiana University School of Medicine, Indianapolis) was cloned in the lentiviral vector pDEST-DH1 with the Gateway system. After lentivirus production with pDEST-DH1-FLAG–Mdm2 vector, the pVSVG plasmid and the psPAX2 plasmid in HEK-293 T, Mdm2-KO MEF cells were transduced with virus-containing supernatants supplemented with 8 μg/mL polybrene and selected with puromycin for stable Mdm2 expression. For cell synchronization at G1/S transition, subconfluent cells were cultured in the presence of 2 mM thymidine for 12 h, and then culture medium for 12 h followed by incubation in the presence of 5 μg/mL aphidicolin for 12 h before release into fresh medium for the times indicated. For cellular arrest in the G2 phase, cells were treated with 5μM etoposide (Sigma-Aldrich, St. Louis, MO, USA) for 18 h and processed for subsequent experimental analysis. For EGFR stimulation, cells were treated with 100 ng/mL EGF (Sigma-Aldrich) after the release of the thymidine-aphidicolin double block for the indicated periods. For blocking Mdm2 activity, cells were treated with the specific catalytic inhibitor HLI737 (3μM, Tocris Bioscience, Bristol, UK) 2 h before the release of the thymidine-aphidicolin double block and maintained thereafter for the indicated times.

### 2.2. Immunoprecipitation and Western Blot

Cell pellets were lysed in RIPA buffer (Tris-HCl 20 mM pH 7.5, 150 mM NaCl, 1% Triton-X100, 0.1% SDS, 0.5% sodium deoxycholate, cocktail of protease and phosphatase inhibitors) for 1 h at 4C and then centrifuged (15,000× *g*, 10 min). 15–30 µg of cleared lysates were resolved in 7.5–13% SDS–PAGE and transferred to nitrocellulose membranes. Membranes were incubated with the indicated primary antibodies: α-actin (1:2000, Santa Cruz Biotechnology, Dallas, TX, USA), GAPDH (1:2000, Santa Cruz Biotechnology), GRK2 (1:1000, Santa Cruz Biotechnology), geminin (1:1000, Santa Cruz Biotechnology), p-histone-3 (Ser10) (1:1000, Millipore, Burlington, MA, USA), Pin1 (1:1000, Santa Cruz Biotechnology), FLAG-M5 (1:500, Sigma-Aldrich) p53 (CM5 antibody, 1:1000, Novacastra, Leica Biosystems, Wetzlar, Germany), DJ1 (1:5000, Abcam, Cambridge, UK), p-CDK1 (Tyr15) (1:1000, Cell Signaling Danvers, MA, USA), CDK1 (1:1000, Santa Cruz Biotechnology), α-Tubulin (1:2000, Santa Cruz Biotechnology). Blots were developed using a chemiluminescent method (ECL, Amersham). Band density was quantitated by densitometric laser analysis.

For immunoprecipitation of Pin1 protein, cells were lysed in 500 μL per 100 mm dish of IP lysis buffer (50 mM HEPES pH 7.5, 150 mM NaCl, 1% Triton, 10% Glycerol, 10 mM NaF,1 mM sodium orthovanadate, plus protease inhibitors). Immune complexes were pulldown with specific anti-Pin1 polyclonal antibody (1 µL/mg protein of clarified lysates, Upstate Biotechnology, Lake Placid, NY, USA). Co-immunoprecipitated proteins were resolved in 15% SDS/PAGE, transferred to nitrocellulose membranes, and immunoblotted for GRK2 detection. Stripped blots were probed for Pin1 levels. Quantified GRK2 was normalized to the amount of the immunoprecipitated Pin1 protein, and data were represented as fold over control conditions.

### 2.3. *In Vitro* Ubiquitination Assays

Ubiquitination of GRK2 by Mdm2 was assessed with purified proteins in a reconstituted In vitro ubiquitination system. Reactions were performed in 50 µL final volume containing 350 ng of recombinant GRK2 protein, 100 ng of purified E1 (Affinity BioReagents, Golden, CO, USA), 200 ng of purified UbcH5b (Boston Biochem, Cambridge, MA, USA), 250 ng of GST-Mdm2 and 5 µg of wild-type ubiquitin or different mutants (lysine-less (no-K), nonconjugable (AA), linkage deficient K48R or triple 3KTR (K29R/K48R/K63R) (Boston Biochem)). Mixtures were incubated for 2 h at 37 °C, and the reaction was stopped by adding an equal volume of 2X nonreducing Lemmli sample buffer supplemented with 1.2 mg/mL N-ethyl-maleimide (Sigma-Aldrich). Ubiquitin-modified proteins were resolved in 7.5% SDS-PAGE and then subjected to Western blot analysis with rabbit polyclonal anti-ubiquitin (1:500, Sigma-Aldrich) and mouse monoclonal (clone C5/1.1, 1:1000, Millipore) or rabbit polyclonal (C-15, Santa Cruz Biotechnologies) anti-GRK2 antibodies.

### 2.4. Polyubiquitinated Protein Pulldown from Synchronized Cells in the G2 Phase

Pulldown of polyubiquitin protein conjugates was performed using agarose Tandem ubiquitin-binding entities, TUBEs (Lifesensors, UM402) according to the manufacturer’s protocol. Both G2-arrested cells with 5 µM etoposide for 18 h and asynchronously growing cells were treated prior to cellular lysis with 40 µM MG-132 for 1 h. Cells were lysed for 1 h at 4 °C in TUBE lysis buffer (Tris-HCl 50 mM pH 7.5, 0.15 M NaCl, 1 mM EDTA, 1% NP40, 10% Glycerol supplemented with phenanthroline 100x and 50 µM PR-619 from Lifesensors) and protein lysates (~0.5 mg) were incubated with agarose TUBEs for 3 h at 4 °C. Bound proteins were resolved in 7.5% SDS–PAGE in parallel with the unbound fraction for subsequent immunoblot analysis.

### 2.5. Immunofluorescence and Confocal Microscopy

Cells previously seeded in glass covers and treated as required for each experiment were rinsed in PBS, fixed in 4% paraformaldehyde (PFA) in PBS for 15 min and permeabilized with ice-chilled methanol for 5 min at −20 °C. Nonspecific sites were blocked by incubation in PBS containing 1% BSA for 1 h at RT. Cells were then washed and incubated over-night at 4 °C with primary antibodies: α-tubulin (1:500, Santa Cruz Biotechnology) and/or pericentrin (1:500, Abcam), followed by incubation for 1 h RT with fluorescent secondary antibodies, and for 5 min with DAPI (1 µg/mL, Merck) to stain nuclei. Images were acquired using a confocal laser microscope LSM710 (Zeiss). Acquisition settings (objective, laser intensities, stacks distance) were maintained in each experiment to allow reliable quantification analysis. The quantitative analysis of acquired cells (objectives, 40X or 63X) was performed using the public domain software for image analysis ImageJ (NIH, Bethesda, Maryland, USA). To measure inter-centrosome distances in the late G2 phase, duplicated centrosomes were identified according to pericentrin signals, and distance was determined by ImageJ. Spindle length was defined as the pole-to-pole distance, where the pole is the position for microtubules convergence in composites of Z-stack images (0.5 µm apart). Cells with condensed chromosomes aligned in a middle plane of the Z-axis of the cell and in between of poles with less than 25 mm apart were scored as metaphases. The extent of chromosome misalignment was quantified by calculating the DNA spread parallel to the spindle pole axis, as previously reported [38].

### 2.6. Kinase Activity Assays

Wild-type and S670A full-length GRK2 were purified from baculovirus-infected Sf9 cells as described [39], and purified N-terminal His6-tagged recombinant human MST2 was from Sigma-Aldrich. Circa 50 nM MST2 protein was incubated with or without 50 nM purified GRK2 proteins for 1–10 min at 30 °C in kinase buffer (20 mM Tris-HCl, pH 7.5, 2 mM EDTA, 7.5 mM MgCl2, 100 μM ATP) in the presence of [γ-32P]ATP (250–1000 cpm/pmol ATP). The reaction was stopped by the addition of SDS sample buffer, and phosphorylated proteins were resolved by 8% SDS–PAGE, stained with Coomassie and visualized by autoradiography for 4 to 24 h. Phosphorylation intensities and levels of Coomassie-stained MST2 and GRK2 proteins were quantified by densitometry, and band densities of 32P-MST2 were normalized to total MST2 values and corrected by GRK2 levels.

### 2.7. Statistical Analysis

Data analysis was performed using GraphPad Prism for Windows. Means between groups were compared with two-tailed unpaired Student’s *t-*test unless otherwise indicated in Figure legends. All results are expressed as means ± SEM.

## 3. Results

### 3.1. Mdm2 Mediates GRK2 Down-Modulation in the G2 Phase

Previous studies from our group showed that GRK2 levels markedly decay during the G2/M and rapidly recover thereafter [31]. During G2, phosphorylation of GRK2 at Ser670 by the CDK2–cyclin A kinase promotes its interaction with the prolyl-isomerase Pin1 and proteasomal degradation of GRK2. This phosphorylation-dependent GRK2/Pin1 association may be critical in recruiting and enabling ligase-mediated ubiquitination for an efficient GRK2 downregulation. Several ligases have been identified that target GRK2 for degradation in different cellular contexts but not during cell cycle progression. In this regard, we have described that the E3 ligase Mdm2 is involved in the ubiquitination and degradation of GRK2 upon GPCR stimulation [33], and, more interestingly, expression of a mutant unable to be phosphorylated at S670 (GRK2-S670A) completely prevents GRK2 down-modulation by Mdm2 in such context [34].

Therefore, we sought to address whether regulation of GRK2 by Mdm2 could also apply to the G2 phase. Total GRK2 levels were analyzed in both wild-type, and Mdm2/p53 double-null (hereinafter referred to as Mdm2-KO) embryonic fibroblasts (MEFs) treated with etoposide, a topoisomerase II inhibitor that causes cell cycle arrest in the G2. As shown in Figure 1A, the down-modulation of endogenous GRK2 protein that takes place in the G2 during cell cycle progression in wild-type cells was abrogated in MEF cells lacking Mdm2, indicating that Mdm2 is required for GRK2 degradation in this context. Levels of geminin, a protein that accumulated from the S phase to the G2 and maintained during mitosis in which it shows a phosphorylation-dependent band shift [40,41], were similarly increased without mobility changes in both Mdm2-KO and wild-type MEFs etoposide-treated cells compared to asynchronously cycling cells (Figure 1A), pointing that differences in GRK2 levels do not result from differences in cell cycle arrest in the G2, but instead from different Mdm2 status.

To better characterize this process and to rule out potential genotoxic effects of etoposide treatment, GRK2 protein levels were analyzed in wild-type or Mdm2-KO MEFs arrested at G1/S and then released synchronously into the cell cycle. The timely decay of GRK2 in the G2 observed in control cells was prevented in the absence of Mdm2 (Figure 1B). Moreover, the ability of Mdm2 to target GRK2 is specifically restricted to the G2 phase as GRK2 is similarly downregulated in both wild-type and Mdm2-KO MEFs at the onset of mitosis, pointing to additional Mdm2-independent processes in this cell cycle phase. The impaired GRK2 degradation in the G2 is not caused by a delayed progression of the Mdm2-KO MEFs into this particular phase compared to wild-type cells, as no differences were observed in profiles of cell cycle markers p-histone 3 and p-Cdk1 (Figure 1B).

Next, we explored whether Mdm2-dependent decay of GRK2 in the G2 relied on S670 phosphorylation, a posttranslational modification involved in GPCR-induced degradation of the kinase by Mdm2 [33,34]. The GRK2-S670A mutant displayed impaired down-modulation in the G2-arrested HeLa cells, while both endogenous and extra wild-type GRK2 was degraded (Figure 2A), in line with previous results in these cells dynamically progressing through the G2 [31]. In addition, the interaction of endogenous Pin1 with GRK2 was increased upon G2 arrest with etoposide in both parental and HeLa cells with extra wild-type GRK2, but not with GRK2-S670A mutant (Figure 2B), suggesting that isomerization of preceding prolyl-peptide bond to phosphorylated S670 can be critical for either Mdm2 recruitment to the kinase or efficient ubiquitination of key lysine residues for degradation. Of note, we have reported that the binding of Mdm2 to GRK2 does not depend on the phosphorylation status of S670 nor subsequent events mediated by Pin1 [34]. Thus, we sought to explore whether GRK2 ubiquitination in the G2 was affected by this regulatory phosphorylation site by using affinity matrix ubiquitin traps called tandem ubiquitin-binding entities (TUBEs), which bind with high-affinity and selectively-polyubiquitinated substrates. Endogenously polyubiquitinated proteins from cellular extracts were pulled down and probed for the presence of GRK2 (Figure 2C). Polyubiquitinated GRK2 was markedly increased in the G2 synchronized HeLa cells that express extra wild-type GRK2 compared to asynchronous cells. Conversely, polyubiquitination of GRK2-S670A mutant was barely detected in the G2 despite similar levels of bound polyubiquitinated proteins in both HeLa-WT5 and HeLa-A1 cells pointing out Mdm2 fails to ubiquitinate mutant GRK2 protein.

Next, we further explored in detail GRK2 ubiquitination by Mdm2 In vitro and compared the ubiquitination patterns obtained for the wild-type kinase, the phospho-defective mutant GRK2-S670A and the catalytically inactive mutant GRK2-K220R, which is poorly phosphorylated on S670 [30,42] and akin to GRK2-S670A is not down-modulated in the G2 [31]. A complex and heterogeneous pattern of GRK2 ubiquitination was detected in the presence of Mdm2 (Figure 2D,E). Both poly- and mono-ubiquitination signals are strongly reduced with a conjugation-defective ubiquitin (ubiquitin-AA), while a lysine-less ubiquitin (a mutant unable to form polyubiquitin chains) allows GRK2 mono-ubiquitination (~80–90kDa size signals) and multi-mono-ubiquitination (~115 kDa size band) (Figure 2D). Interestingly, Mdm2 also triggers polyubiquitin chains of different linkages and topology as ubiquitinated GRK2 species of higher molecular weight are dependent on ubiquitin attachment via K29 and/or K63 linkages, but those of lower weight (~128 kDa size band) persist in the absence of lysine 29, 48 and 63 linkages sites, as shown with the use of ubiquitin mutants differentially altered in these sites (Figure 2E). Of note, the GRK2-S670A and K220R mutants showed different ubiquitination patterns compared to WT, and particularly levels of highly poly-ubiquitinated GRK2 were severely reduced, in line with a less efficient proteasome targeting and degradation. Conversely, neither Mdm2-induced mono-ubiquitination nor multi-monoubiquitination of GRK2 was affected by phosphorylation on S670 (Figure 2D), suggesting these modifications either control kinase turnover uncoupled from cell-cycle or additional functions unrelated to protein stability. Overall, these data support that following S670 phosphorylation and likely Pin1-mediated prolyl-isomerization, the formation of heterogeneously linked polyubiquitin chains by Mdm2 may accelerate the turnover of GRK2 in the G2.

### 3.2. Mdm2 Regulates Centrosome Separation in the G2 Phase

Ubiquitin and proteasome-mediated degradation of cell cycle regulators is a widespread strategy to push cells through the cell division cycle and coordinate the timing and strength of key events in the centrosome cycle [15,16]. We had previously shown that halt of GRK2 degradation occurring in the transition of the G2 phase into mitosis is part of a DNA checkpoint mechanism ensuring DNA integrity [31]. In addition to allowing DNA control before chromosome segregation, G2 accomplishes other essential processes, such as maturation and separation of centrosomes for positioning of the mitotic spindle [4,5,6], and a role for GRK2 in mitogen-promoted centrosome separation has been reported [23]. Therefore, we sought to address whether Mdm2-dependent degradation of GRK2 played a regulatory role in centrosome disjunction. Centrosome separation was measured in both Mdm2-KO MEFs and derived cells with stable reexpression of Mdm2 (Mdm2 knock-in (KI) MEFs, Figure 3A). In cells synchronously progressing in the G2 for 6 h after release from G1/S arrest (Figure 3B), inter-centrosomal distances were markedly higher in the absence (17.45 ± 0.47 µm means ± SEM, n = 106) compared to the restored presence (8.95 ± 0.24 µm, means ± SEM, n = 100) of Mdm2. Interestingly, these lower distances seem to correlate with the recovery of timely downmodulation of GRK2 in the G2 linked to Mdm2 reexpression (Figure 3C) and not to differences in cell cycle progression as cell cycle markers displayed similar profiles. As compared to Mdm2-KO MEFs, the range of inter-centrosomal distances in Mdm2-KI MEFs is similar to that reported in MEFs of wild-type genetic background [43,44], pointing that lack of p53, which presence negatively controls aurora-A-mediated centrosome separation [45], may play a minor role in impairing inter-centrosomal cohesion in our experimental model.

Next, to further document the involvement of Mdm2-mediated GRK2 turnover in centrosome separation in cells that are synchronously progressing in the G2 with endogenous expression of the ligase, we analyzed the effect of the ligase inhibitor HLI373 on mitogen-promoted centrosome disjunction, a process known to be positively influenced by GRK2 in the well-characterized cellular setting of HeLa cells [23]. Using the reported 2 microns of the normal distance between centrioles as representing non-separated centrosomes [46], we observed a similar percentage of cells with centrosomal separation after EGF treatment in control and Mdm2-inhibited conditions (Figure 4A), suggesting that inhibition of Mmd2 activity in the G2 had no major effects on cell cycle progression nor EGF-dependent onset of centrosome disjunction. However, as shown in Figure 4B, inhibition of the Mdm2 ligase activity in synchronously progressing cells into the G2 for 6 h in the presence of EGF caused a statistically significant increase in the average inter-distance of separated centrosomes (8.35 ± 0.62 µm, means ± SEM, n = 114 in control cells versus 11.43 ± 0.91 µm, means ± SEM, n = 96 in Mdm2-inhibited HeLa cells, p = 0.004 in unpaired Student’s *t*-test). Previous studies had reported that median centrosomal separation in control HeLa cells reached 9 µm in late G2/prophase, occasionally up to 12–15 µm [44,47]. Thus, we stratified cells according to inter-centrosome distances that were ranged in non-separated (0–2 µm), separated one standard deviation of the mean (>2–15 µm) or separated over 15 µm (15+). Interestingly, the percentage of cells with centrosomes migrating distances over 15 µm was increased 3-fold after 6 h of release from G1/S arrest in the presence of the HLI373 inhibitor (Figure 4C). Of note, GRK2 was readily downmodulated in cells progressing into the G2 with the presence of EGF, but not when Mdm2 is inhibited (Figure 4D). Collectively, these data are consistent with a specific effect of HLI373 in protracting the EGFR pathway of centrosome separation through the abrogation of Mdm2-mediated turnover of GRK2, which leads to a higher distance between centrosomes in late G2.

### 3.3. GRK2-Associated Defects in Centrosome Separation Correlate with Altered Spindle Lengths and Chromosome Congression

We have established in different cellular contexts that phosphorylation of S670 downstream diverse signaling cascades is a key event enabling GRK2 degradation by Mdm2. Interestingly, EGF stimulates centrosome separation but also triggers MAPK-dependent phosphorylation of GRK2 in many settings (reviewed in [25,26]). Hence, we next sought to evaluate the effect of preventing this posttranslational modification in EGF-induced centrosome separation in late G2. Using synchronized parental cells or stable HeLa cells overexpressing wild-type GRK2 (HeLa-WT5) or the non-phosphorylatable GRK2-S670A mutant (HeLa-A1), we observed that duplicated centrosomes separation in EGF-treated cells correlates with steady-state-levels of GRK2 as long as its degradation is normally allowed in the G2. Thus, separation distances with extra amounts of degradation-competent wild-type GRK2 (11.61 µm ± 0.757, n = 31) were higher than those in parental cells (9.631 µm ± 0.4499, n = 31) (Figure 5A), suggesting that excess GRK2 may not be degraded by Mdm2 so efficiently as to reduce protein levels to a threshold that ensures optimal centrosome separation. Surprisingly, expression of GRK2-S670A, an Mdm2-dependent degradation-defective protein, markedly impaired centrosome separation in the G2 phase, despite kinase protein levels remained high and were similar to those observed upon wt GRK2 stabilization in HLI373-treated HeLa cells, which displayed enhanced separation instead (see Figure 4C). Since cells with extra GRK2-S670A are normally progressing in the cell cycle after 6 h of release from G1/S transition (Figure 5B), we reasoned that the observed discrepancy with data obtained in HeLa-WT5 cells and HLI373-treated parental cells might reflect an intrinsic incapacity of GRK2-S670A protein to phosphorylate MST2. In keeping with this, we have demonstrated that GRK2-S670 phosphorylation promotes a switch in GRK2 substrate specificity as this modification at the C-terminus may influence allosteric kinase activation [42,48]. Hence, we performed In vitro phosphorylation assays using purified glutathione S-transferase (GST)–MST2 in the presence of recombinant wild-type GRK2 or GRK2-S670A proteins (Figure 5C). Our data showed significant autophosphorylation of MST2 over the time-course analyzed and increased MST2 phosphorylation catalyzed by wild-type GRK2 as previously reported [23]. Interestingly, GRK2-S670A failed to increase phosphorylation over the MST2-autophosphorylation signal despite this kinase mutant is competent to modify other well-known GRK2 targets [40,46], pointing that MST2 is an EGF-dependent biased substrate of GRK2 as previously proposed for HDAC6 [49]. Indeed, we have suggested that phosphorylation of GRK2 on S670, a residue located in a region of the protein that allosterically communicates with the kinase catalytic domain, may facilitate a slightly different conformation of the active site, which may enable catalysis in a specific substrate-dependent (“biased”) manner. Therefore, impaired centrosome separation in HeLa-A1 cells would result from the stabilization of a GRK2 protein unable to activate MST2 in the G2 and trigger EGF-stimulated centrosome tether cleavage.

EGF-induced phosphorylation of S670 catalytically activates GRK2 towards MST2 to initiate centrosomal disjoining but also enables a phosphodegron that signals the degradation of GRK2 by Mdm2. Therefore, the kinase axis GRK2/MST2/Nek2A would be counterbalanced by the same event that causes its stimulation. We reasoned that the equilibrium of these opposing processes might be important for ensuring proper positioning and length of the mitotic spindle, which is key to guarantee precise attachment of the spindle microtubules with the kinetochores of sister chromatids. Hence, we measured interpolar distances of metaphase spindle, which maintains constant dimensions despite the microtubules undergo intensive dynamics, in cells with extra (HeLa-WT5) or defective (HeLa-A1) GRK2 catalytic activity towards MST2 (Figure 5D). Compared to HeLa parental cells, we observed significantly shorter spindle lengths in the presence of GRK2-S670A, in line with its impaired centrosome disjunction in the G2 (Figure 5A). Conversely, interpolar distances were higher in HeLa-WT5 cells. Next, we analyzed whether deviations from normal length spindle could have an impact on chromosome alignment. Interestingly, there is a trend for chromosome congression errors in both HeLa-A1 and WT5 cells (Figure 5D), which could be a result of the altered intercentrosomal distances noted in these cells.

Overall, these results point out that GRK2 fosters centrosome separation by means of its participation in a complex regulatory loop coordinated by EGF-induced phosphorylation of this kinase at S670.

## 4. Discussion

Different routes of centrosome separation support the assembly of the bipolar spindle, which final geometry, length and orientation have significant consequences in chromosome segregation [1,2,5,13,14]. Despite cumulative knowledge about mechanisms of separation and molecular players, the cues for cell type-specific timing of centrosome disjunction and optimal separation distance are only beginning to be deciphered. Recently, growth factors have emerged as regulatory determinants of the timing of centrosome disjunction. The addition of EGF initiates premature separation of centrosomes in the G2 [22], and GRK2 kinase mediates this process by means of phosphorylation and activation of MST2 [23], which in turn stimulates Nek2A, a central kinase in dispersing the centrosomal linker [8,9,10,11]. Our results show now that EGF-induced phosphorylation of GRK2 on S670 is a key event in initiating centrosome separation and also a relevant clue for limiting centrosome separation, as this event simultaneously triggers the Mdm2-dependent ubiquitination and degradation of GRK2 (Appendix A).

Several “cell-cycle driver” kinases have been implicated in centrosome dynamics [12]. PLK1 promotes disjunction by activating MST2 and prevents the association of MST2/Nek2A with PP1γ, a phosphatase that counteracts Nek2A activity at centrosomes [20]. CDK1 is reported to inhibit PP1γ and activate the aurora A/PLK1 module [45] for the stimulation of NIMA kinases Nek9/Nek6/Nek7 cascade that phosphorylate Eg5, the main motor protein in centrosome movement [47]. In addition, the activity of CDK2 in the G2 is involved in the coordination of centrosome duplication with the cell cycle [4,6] and less clearly in centrosome separation by coordinating the activation of CDK1 [50]. Our findings are consistent with GRK2 being an intrinsic regulatory component of the centrosome cycle machinery via CDK2 as an upstream activating kinase for subsequent GRK2 phosphorylation of MST2 (Appendix A). Interestingly, GRK2 is a substrate of the CDK2–cyclin A complex In vitro and in a cellular context, with residue S670 being the main phospho-acceptor site [31]. Both CDK2–cyclin A and GRK2 localize to the centrosome in the G2 phase [23,50]. Moreover, phosphorylation of GRK2 at S670 parallels activation of MST2-kinase activity and Nek2A functionality, which peak in the G2/M phase [21,31,51]. Altogether, it is tempting to suggest that CDK2-cyclin A phosphorylation of GRK2 and MST2 activation might define an additional route of centrosome disjunction paralleling the well-described PLK1-MST2–Nek2A pathway, wherein mitogenic signals can converge through S670-GRK2 phosphorylation.

### 4.1. The Multifaceted Mdm2 Joins the Control of Centrosome Separation

Centrosome cycling is tightly controlled by the degradation of centrosome-associated factors. For instance, Nek2A is actively degraded to prevent that centrosomes split out the G2/M window [15], and PLK4 levels decline at S-phase to impede reduplication of centrioles [16]. Our data reveal that GRK2 is subject to active degradation by Mdm2 to curtail activation of the MST2/Nek2A pathway and limiting centrosome separation. Mechanistically, both MAPK and CDK2 can drive the phosphorylation of GRK2 on S670, allowing Pin1 recruitment and Mdm2-mediated ubiquitination of GRK2 protein. In line with our data identifying Mdm2 as the GRK2’s ligase in the G2, Mdm2 and GRK2 protein levels fluctuate through S and G2/M in opposite ways [31,52]. Interestingly, Mdm2 becomes phosphorylated by CDK2–cyclin A during S-phase progression and G2, switching its repertoire of substrates [53]. It is tempting to suggest that CDK2 would simultaneously phosphorylate Mdm2 and GRK2 to favor the timely degradation of GRK2 in the G2 phase. Indeed, GRK2 mutants with deficient phosphorylation in this residue (S670A and K220R) are unable to be downmodulated during the G2 [31] and show different ubiquitination patterns compared to WT (Figure 2). Interestingly, Mdm2 seems to promote different patterns of modifications on GRK2 ([33] and the present work), which may confer different regulatory features. Mdm2-induced mono-ubiquitination of GRK2 is not affected by the status of S670, but in the absence of S670 phosphorylation, Mdm2 is unable to trigger polyubiquitination and turnover of GRK2 protein. Polyubiquitin-tagged proteins are recognized and degraded by the proteasome, although with different efficiencies depending on the ubiquitin chain length and topology. Mdm2 can form polyubiquitin chains containing all possible isopeptide linkages [54], but we find a preferential assembly of chains on GRK2 containing K29- and/or K63-linkages, similar to other substrates in which the ligase mainly utilizes Lys11, Lys29 and Lys63 for ubiquitination [55]. Interestingly, heterogeneous chains formed with mixed isopeptide linkages are recognized by the proteasome even more efficiently than canonical K48-linked chains [56]. Therefore, it is tempting to suggest that polyubiquitination of GRK2 by Mdm2 is a hierarchical process guided by the extent of GRK2 phosphorylation on S670 and Pin1 binding. The pSer670-mediated GRK2-Pin1 association could allosterically unmask Mdm2 recognition motifs on GRK2 to facilitate their interaction or make accessible key lysine residues for more efficient ubiquitination patterns.

Cumulative evidence points out centrosomes as scaffolds for proteasomal degradation in the regulation of diverse cellular processes [15]. Active proteasomes and ubiquitin ligase complexes at centrosomes support the degradation of both resident centrosomal and dynamically gathered extra-centrosomal proteins without markedly reducing their overall cellular abundance, which may be particularly advantageous for proteins with catalytic functions and ubiquitous subcellular distribution, such as GRK2. This local degradation would allow higher reductions of GRK2 protein, readily affecting the pool of MST2 localized in centrosomes. Of note, GRK2 protein is found in the centrioles and pericentriolar region of centrosomes [23], and Mdm2 activity is linked to centrosome functionality [38]. It is tempting to suggest that GRK2 downmodulation in the G2 is circumscribed to centrosomes, while kinase functions in other locations are not affected.

Our work reveals an unforeseen function of Mdm2 in the regulation of early disjunction and prophase separation of centrosomes, which may be part of its oncogenic activity. As the negative regulator of p53, overexpression of Mdm2 is known to affect genomic instability since p53 deficiency leads to reduced DNA repair activity [57]. P53 also regulates centrosome cohesion by restraining the activity of the aurora A/PLK1 module [45]. Hence, extra levels of Mdm2 could indirectly promote centrosome separation via p53, but the reduction of inter-centrosomal distances in the absence of p53 (Figure 3) argues for a direct role of Mdm2 in this process. Our results point out that Mdm2-mediated degradation of GRK2 in late G2 is intended to reduce the activity over MST2/Nek2A pathway, preventing excessive centrosome separation and abnormal chromosome congression (Appendix A). This role of Mdm2 would be consistent with suggested tumor suppressor functions proposed for this ligase in the appropriate context [57]. Moreover, it may also be inferred that both excessive and defective Mdm2 activity towards GRK2 would mean deviations from optimal centrosome separation in prophase and represent a risk for proper chromosome alignment. In line with this duality, Mdm2 is associated with a better or worse prognosis depending on the tumor type [58].

### 4.2. GRK2 Fluctuates to Set Centrosome Separation and Spindle Length

Migration of split centrosomes in prophase relies on the activity of CDK1/Nek9/Nek6-7/Eg5 pathway and microtubules, which also cooperate in centrosome disjunction [8,9,10]. The motor protein Eg5 can generate the force required for breaking the centrosomal linker by sliding antiparallel MTs in opposite directions, thereby pushing centrosomes apart at the mitosis onset. Therefore, in the absence of MST2/Nek2A activity, normal bipolar spindles of proper length are still generated whenever the Eg5 pathway was operative [21]. The acetylation state of tubulin plays a relevant role in the recruitment of motor protein Eg5 to MTs. Tubulin hypoacetylation due to dysregulation of negative regulators of the cytosolic deacetylase HDAC6 (histone deacetylase 6) abrogates Eg5-dependent separation of centrosomes, increasing the appearance of monopolar spindles [59]. Interestingly, prior phosphorylation of GRK2 at S670 was required for allowing activation of HDAC6 and α-tubulin deacetylation, leading to increased MT instability, while GRK2-S670A mutant has the opposite effect [48]. It is tempting to suggest that the presence of this mutant could foster the Eg5 pathway of centrosome separation, partially compensating for the defects in the MST2/Nek2A pathway and allowing the assembly of bipolar spindles.

Spindle length can be influenced by microtubule dynamics and mitotic motor activity [3,5], and unbalanced MT depolymerization results in shorter spindles [60]. Intriguingly, our results show that GRK2 mutants impairing MST2 phosphorylation also promote abnormal lengths of the metaphase spindle. It is possible that besides centrosome disjunction GRK2 could affect the Eg5-microtubule-motor pathway in early mitosis via MST2–Nek2A. Nek2 antagonizes the microtubule-stabilizing activity of centrosome-associated factors during interphase [61]. Alternatively, GRK2 could directly modulate MT stability, thereby influencing spindle axis length independently of MST2/Nek2A. Activation of HDAC6 by GRK2 increased MT instability, while GRK2-S670A mutant has the opposite effect [48]. Akin to cells expressing GRK2-S670A protein, reduced spindle length and misaligned chromosomes have also been associated with MT hyperacetylation by HDAC6 inhibition [62]. Alternatively, scaffolding functions of HDAC6 could be altered by mutated GRK2. In this regard, HDAC6 interacts with the plus-end tracking protein EB1, which depletion shortens spindle length [60]. Although further studies are needed to discriminate between these possibilities, fine-tuned control of GRK2 levels by Mdm2 emerges as a relevant process in centrosome separation and spindle length.

Our data reinforce the notion that timely induction and extent of MST2/Nek2A activity must be tightly regulated to render properly separated centrosomes in mitosis. Different studies have shown that sufficient separation of centrosomes in advance to NE breakdown is advantageous for cells [13,14], as this event favors faster assembly of metaphase spindle and improves chromosome alignment in metaphase. In agreement with an increase in Nek2A activity [11], excess of GRK2 levels, as a result of either increased transcription (heterologous cDNA overexpression) or protein stabilization (small molecule inhibitors/knockout expression of Mmd2), converging with increased S670 phosphorylation (linked to hyperactivation of mitogenic pathways) would cause centrosomes to move further apart in the G2. Conversely, impaired phosphorylation of MST2 by GRK2-S670A strongly delays centrosome splitting and movement in the G2, consistently with a defective Nek2A activation [11]. Furthermore, both insufficient and premature separation of centrosomes before NE breakdown can be a source of aberrant kinetochore attachments and chromosome segregation errors that may contribute to aneuploidy and tumorigenesis [14,43]. In line with this, the extent of chromosome misalignment in the metaphase plate significantly increases in HeLa cells with extra wild-type GRK2 and mutant GRK2-S670A (Appendix A). Therefore, both up and down GRK2 levels may converge in chromosome segregation errors. It is worthy of note that both types of GRK2 dysfunction are frequent in several cancer types [26]. Such changes play tumor-type specific roles in cancer progression by fostering proliferation, survival, or invasive motility [24,25,26]. Our results point out that centrosome and spindle alterations could be a novel pro-tumoral effect of GRK2 instructed by both gain and loss of protein kinase levels in different types of cancer. A proper balance of outcomes of Ser670-phosphorylated GRK2, simultaneously affecting prophase centrosome separation via activation of the MST2/Nek2A axis and targeting GRK2 to Mdm2-mediated degradation pathway, may be a critical factor in genome stability.

## Figures and Tables

**Figure 1 cells-10-00729-f001:**
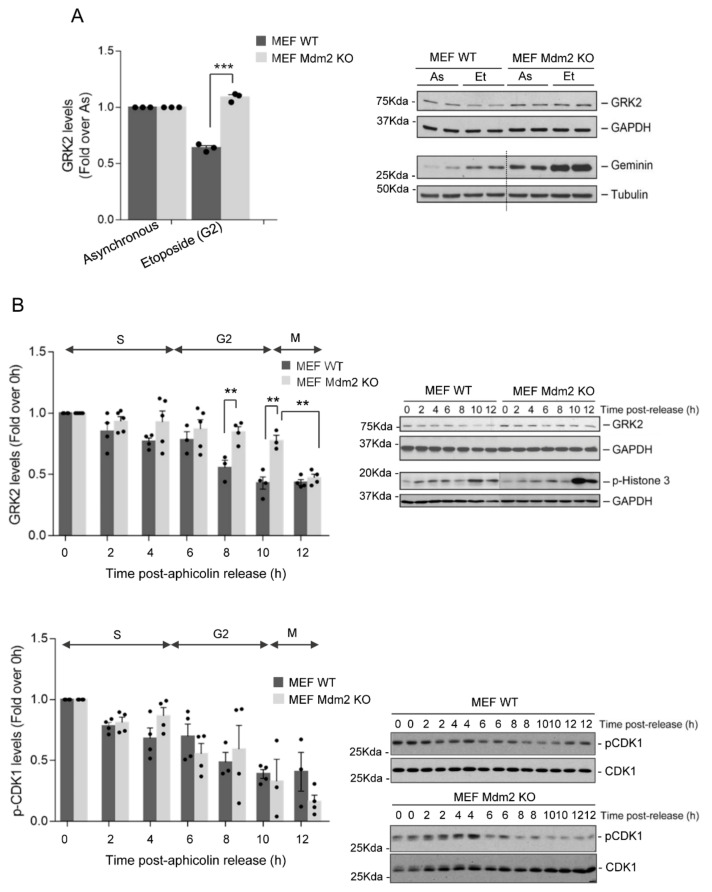
Protein decay of GRK2 in the G2 phase of the cell cycle depends on Mdm2. (**A**) Down-modulation of endogenous GRK2 in the G2 is abrogated in cells lacking Mdm2. Wild-type (WT) and Mdm2-null (KO) mouse embryo fibroblasts (MEF) cells were maintained in exponentially growing conditions (asynchronous, As) or synchronized in the G2 phase by treatment with 5μM etoposide (Et) for 18 h. GRK2 levels in RIPA buffer cell lysates were analyzed by immunoblotting and normalized with tubulin as a loading control. Geminin accumulation in the G2 was used as a synchronization marker. Dotted lines indicate cropped lanes from the same blot. (**B**) Both WT and Mdm2-KO MEF cells were arrested in G1/S transition with a thymidine-aphidicolin double block as detailed in Methods and released into the cell cycle for the indicated times. Protein levels of GRK2, pSer10-histone 3 as cell cycle marker, and GAPDH as loading control were analyzed with specific antibodies (upper panel). Cell cycle progression through the G2 phase is not affected by the loss of Mdm2 in MEF cells (bottom panel). Activation of CDK1 was monitored as a decrease in the extent of the inhibitory Y15 phosphorylation. Levels of pTyr15-Cdk1 were detected with a specific antibody, and blots were re-probed after stripping with anti-CDK1 antibody. Data are means ± SEM from 3–4 independent experiments. Representative blots are shown. (** *p* < 0.01, *** *p* < 0.001, two-tailed *t*-test). Detailed information about the Western blots can be found in Appendix A.

**Figure 2 cells-10-00729-f002:**
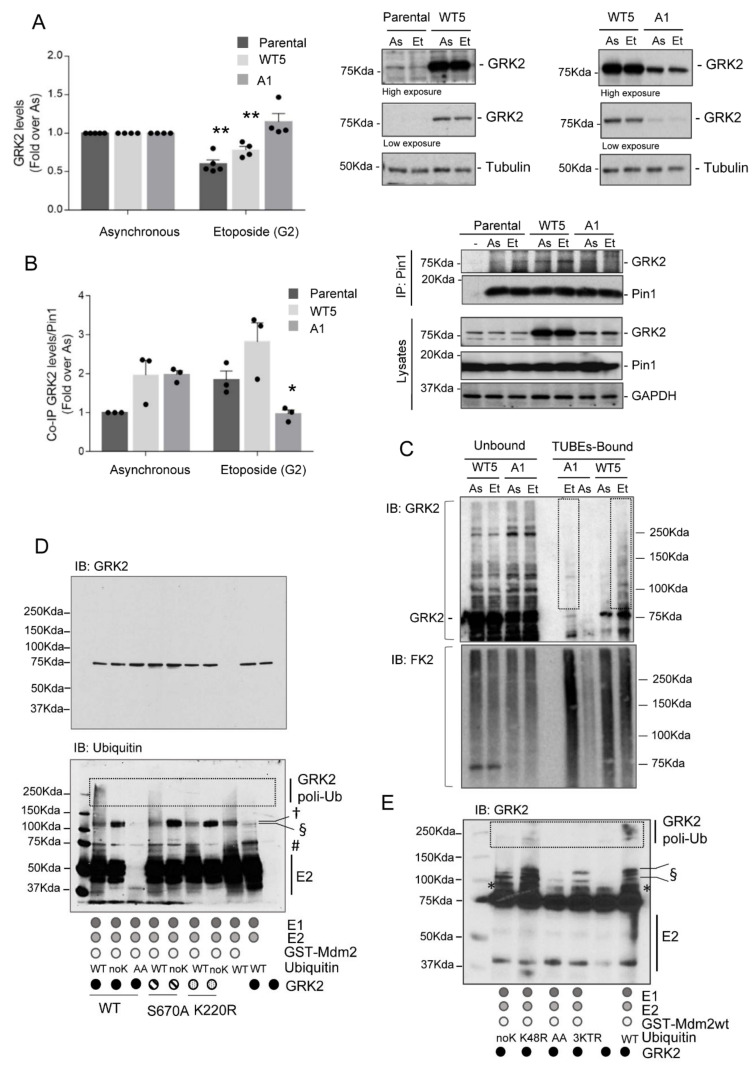
Phosphorylation on Ser670 is required for Mdm2-mediated ubiquitination and down-modulation of GRK2 in the G2. (**A**) Lack of Ser670 phosphorylation stabilizes GRK2 protein levels in the G2 phase of the cell cycle. HeLa cells stably overexpressing wild-type (HeLa WT5) or mutant GRK2-S670A (HeLa A1) and parental cells were incubated in the absence (asynchronous, As) or presence of etoposide (Et) for the G2 synchronization as in Figure 1. Cells were lysed in RIPA buffer, and total GRK2 and tubulin were analyzed by immunoblotting. GRK2 levels were normalized with tubulin as a control of protein loading. Representative blots are included. Dotted lines indicate that intervening lanes have been spliced out. (**B**) Enhanced interaction of GRK2 with the cell cycle regulator Pin1 relays on Ser670 phosphorylation during the G2. Cellular lysates of asynchronously growing or G2-arrested HeLa WT5, HeLa A1 and parental cells were immunoprecipitated with a specific Pin1 antibody and presence of GRK2 analyzed by immunoblot. Immunoprecipitated Pin1 was immunodetected after membrane stripping and used to normalize Pin1-bound GRK2 levels. Input levels of GRK2 and Pin1 proteins and tubulin as control of loading were determined in whole lysates. Values are means ± SEM from 3-5 independent experiments performed in duplicate (* *p* < 0.05, ** *p* > 0.01). Illustrative blots are shown. (**C**) TUBEs affinity pulldowns were performed on HeLa WT5 and A1 cells cultured for 18h in the absence or presence of 5 µM etoposide for the G2 synchronization. Protein ubiquitination was stabilized with the addition of the proteasome inhibitor MG-132 (40 µM) 1h prior to cellular lysis. Both TUBE-bound polyubiquitinated proteins and unbound proteins were immunoblotted with the specific mouse monoclonal antibody c5/1.1 that detects GRK2 independently of its ubiquitination state, as reported previously [33]. After stripping, blots were re-probed with the anti-ubiquitin antibody FK2, which recognizes only polyubiquitinated proteins, confirming similar trapping by TUBEs in the G2-arrested cells. Representative blots of two experiments are shown. (D,E) In vitro ubiquitination patterns driven by Mdm2 vary with the extent of GRK2 phosphorylation on Ser670. (**D**) Recombinant GRK2-wt, S670A or K220R proteins were incubated in the presence of E1 and E2 protein (UbcH5b), GST-Mdm2 as E3 ligase and different purified ubiquitin proteins (WT, no-K (unable to form polyubiquitin chains into protein targets) and AA (unable to be conjugated to E1 thus blocking any ubiquitination)). Ubiquitination was detected by immunoblotting with an antibody able to recognize both free and pan-conjugated ubiquitin as indicated in Methods. Equal loading of GRK2 protein levels was confirmed by blot re-probing with a rabbit polyclonal antibody (C-15, Santa Cruz) that preferentially recognizes non-ubiquitinated forms of the kinase protein. (**E**) Effect of ubiquitin-chain elongation mutants on GRK2 polyubiquitination induced by Mdm2. Ubiquitination of GRK2-wt was performed as above in the presence of additional ubiquitin mutants that interfere with the assembly of specific polyubiquitin chains (K48R and triple mutant K29R/K48R/K63R (3KTR)). Samples were analyzed by Western blot using mouse monoclonal antibody c5/1.1. †, denotes protein bands in the blot corresponding to E1 or GST-Mdm2-related mono-ubiquitination signal (E1 enzyme, MW~110 KDa; GST-Mdm2, apparent MW~108). §, denotes GRK2-related multi-mono-ubiquitination band. #, denotes GRK2-related mono-ubiquitination band. ***** ubiquitin-unrelated GRK2 signal. Dashed boxes indicate polyubiquitinated proteins. Detailed information about the Western blots can be found in Appendix A.

**Figure 3 cells-10-00729-f003:**
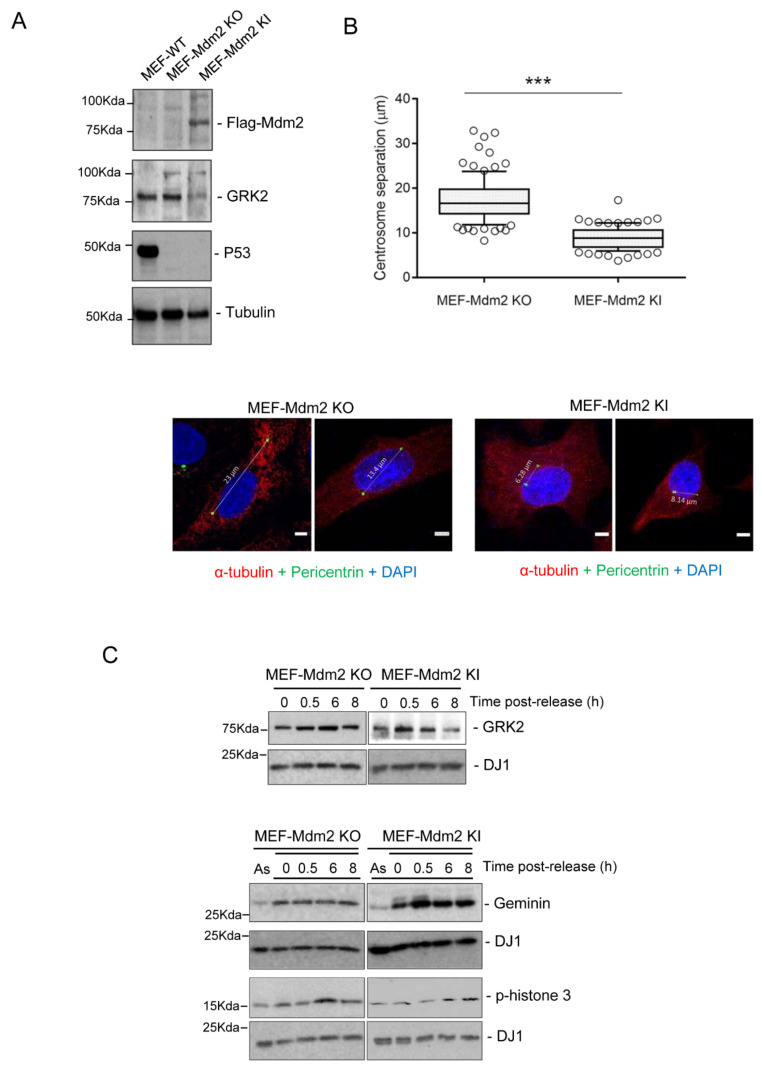
Mdm2 restrains centrosome separation in the G2 phase in parallel to GRK2 downmodulation. (**A**) GRK2 is downregulated by Mdm2 in MEF cells. Ligase activity was restored in p53 and Mdm2 double-knockout MEFs (Mdm2 KO) with lentiviral constructs of Flag-tagged human Mdm2 to generate MEF Mdm2 KI cells. Primary MEFs obtained from C57BL6 mice were used as wild-type control cells. RIPA lysates from exponentially growing cells were analyzed by immunoblot using specific anti-Flag tag, anti-GRK2 and anti-p53 antibodies. Tubulin was used as a control of protein loading. A representative blot is shown. (**B**) Both Mdm2 KO and Mdm2 KI MEF cells were synchronously released into the cell cycle for 6 h after G1/S arrest with a thymidine-aphidicolin double block. Cells were fixed and stained for interphase microtubules with anti-α-tubulin antibody (red), centrosomes with anti-pericentrin antibody (green) and DNA with DAPI (blue). Distances between the two centrosomes of cells were quantified in confocal images. Data are mean± SEM from two independent experiments represented in box-and-whiskers plots: boxes show the upper and lower quartiles (25–75%) with a line at the mean, whiskers extend from the 10 to the 90 percentiles (mean: Mdm2 KO = 17.45 µm, n = 106 cells; Mdm2 KI = 8.95 µm, n = 100 cells). *** *p* < 0.001, two-tailed *t*-test. Representative cells for each condition are shown, scale bar 5 µm. (**C**) Levels of geminin and pSer10-histone 3 confirm that Mdm2 KO and Mdm2 KI MEF cells display similar progression in the G2 after 6 h of post-release from G1/S arrest, as shown by the increase in pSer10-histone 3 in cells without band shift of geminin. Reintroduction of Mdm2 restores timely decay of GRK2 levels in the G2 phase. Immunoblots of RIPA cell lysates were probed with the indicated antibodies. Levels of DJ1 and actin confirm similar protein loading. Representative blots are shown. Detailed information about the Western blots can be found in Appendix A.

**Figure 4 cells-10-00729-f004:**
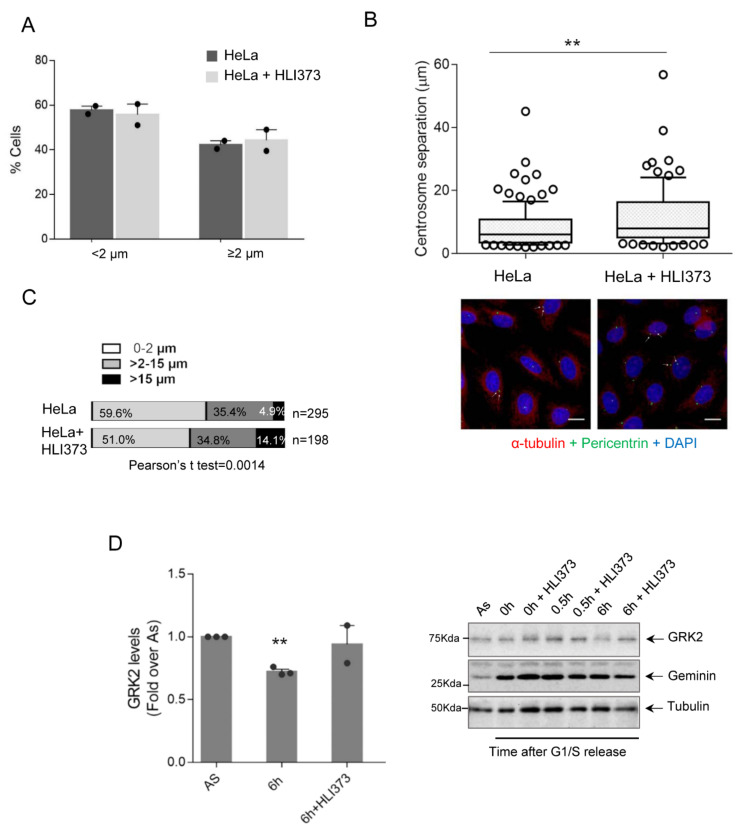
Inhibition of Mdm2-dependent GRK2 degradation in the G2 correlates with enhanced EGF-induced centrosome separation. HeLa cells were synchronized at G1/S transition with a thymidine-aphidicolin double block and released into the cell cycle for 6 h in the presence of 100 ng/mL EGF with or without the Mdm2 inhibitor HLI373 (3 µM). Cells were fixed and co-stained with anti-pericentrin (green) and anti-α-tubulin (red) antibodies in combination with DAPI (blue) for DNA detection. Distance between two duplicated centrosomes was quantified. (**A**) Percentage of cells with separated (≥2 μm) and non-separated (<2 μm) centrosomes. Data are means ± SEM from 2 independent experiments with n > 200 cells per condition in each experiment. (**B**) Inter-centrosomal distances in cells with separated centrosomes in control or HLI373-treated conditions under EGF stimulation. Data are means ± SEM depicted in a box-and-whisker plot (mean: control = 6.1 µm, n = 115; HLI373 = 8.0 µm, n = 97). Confocal images of representative cells are shown. Scale bar, 10 μm. ** *p* < 0.01, Mann–Whitney test. (**C**) Percentage of cell distribution according to non-separated duplicated centrosomes (0–2 μm), centrosomes with separation between 2 and 15 μm and centrosomes separated with larger distance (>15 μm). (**D**) Effect of HLI373 treatment on GRK2 levels. Immunoblots of RIPA lysates from control and HLI373-treated cells in the presence of EGF stimulation and progressing into the G2 for the indicated times after G1/S release were assessed for GRK2 and anti-tubulin as control of protein loading. Geminin levels confirm comparable synchronization in both cellular conditions. ** *p* < 0.01, two-tailed *t*-test. A representative blot is shown. Detailed information about the Western blots can be found in Appendix A.

**Figure 5 cells-10-00729-f005:**
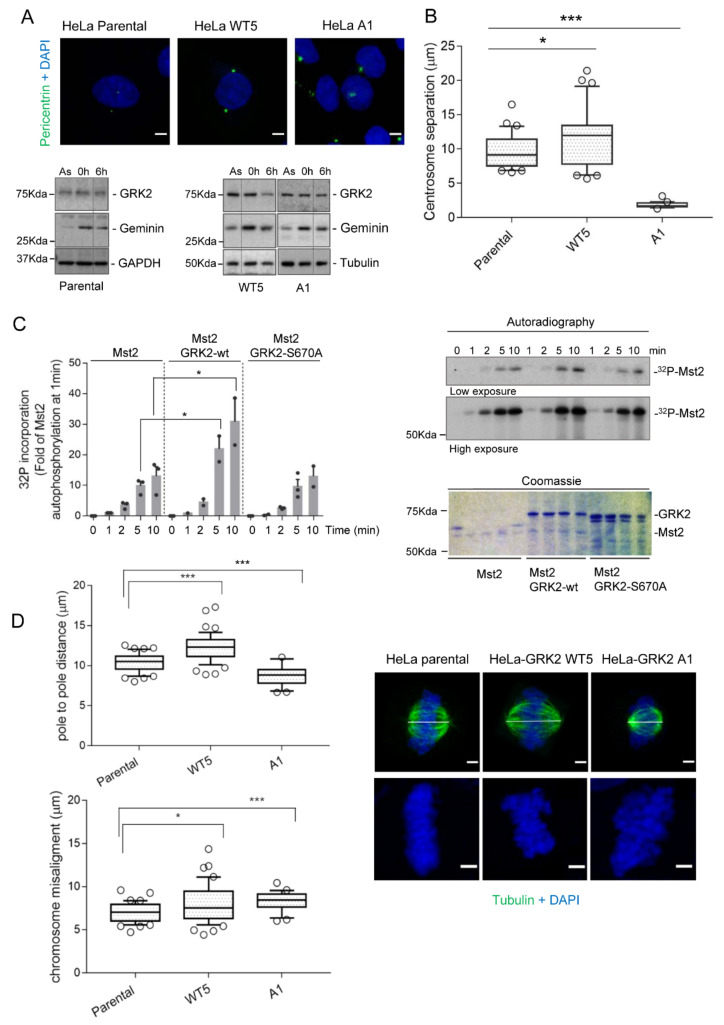
Proper centrosome separation in the G2 and chromosome alignment depends on Ser670 phosphorylation levels balancing Mdm2-mediated decay of protein levels and kinase activity of GRK2 towards MST2. (**A**,**B**) Centrosome separation in the G2 directly correlates with protein levels of catalytically active and Ser670 phosphorylation-enabled GRK2. HeLa cells stable overexpressing wild-type (HeLa WT5) or mutant GRK2-S670A (HeLa A1) and parental cells were synchronized at G1/S transition and released into the cell cycle. After 6 h of progression, cells in late G2 were stained for pericentrin (green) as centrosome marker and DNA (DAPI, blue) (**A**). Representative confocal images are shown. Scale bar, 5 μm. Protein levels of GRK2 and geminin as a marker of similar cell cycle progression from S to the G2 phase were analyzed by Western blot in cells growing asynchronously, arrested at G1/S transition (0 h) or progressing 6 h after release from G1/S arrest. Tubulin and GAPDH levels were used as loading controls. Dotted lines indicate cropped lanes from the same blot. (**B**) Distances between centrosomes are means ± SEM data plotted in a box-and-whisker plot. Mean: parental = 9.6 µm, n = 31 cells; HeLa-Wt5 = 11.6 µm, n = 31 cells; HeLa-A1 = 1.8, µm, n = 21 cells. (**C**) The Ser670 phospho-defective GRK2 mutant displays a markedly reduced ability to phosphorylate MST2. In vitro time course assays of MST2 phosphorylation were performed in the presence of [γ-^32^P]-ATP using recombinant GRK2-WT or GRK2- S670A as described in Materials and Methods. The intensity of ^32^P and the Coomassie bands were quantified by densitometry and analyzed as described in Methods. Results were plotted as fold of MST2-triggered 32P incorporation in the absence of GRK2. Data are the means ± SEM from 2–3 independent experiments. A representative autoradiograph with its corresponding Coomassie staining is shown. (**D**) Cells with unbalanced GRK2 activity towards MST2 show altered lengths of the mitotic spindle and defective chromosome congression. Exponentially growing HeLa WT5, HeLa A1, and parental cells were fixed and stained for microtubules (α-tubulin, green) and DNA (DAPI, blue). The mitotic stage of circa 120 cells in each condition from three replicate experiments was scored according to the distinctive appearance of chromatin at prophase, metaphase, anaphase, and telophase. The pole-to-pole distances of mitotic spindle were quantified in metaphase (mean: parental = 10.4 µm, n = 41 cells; HeLa-WT5 = 12.3 µm, n = 48; HeLa-A1 = 8.7 µm, n = 27). The extent of chromosome misalignment was measured as the maximal distance of DNA spread parallel to the spindle pole axis. Data (means ± SEM) are represented in box-and-whisker plots (mean: parental = 7.0 µm, n = 41 cells; HeLa-WT5 = 7.9 µm, n = 48; HeLa-A1 = 8.3 µm, n = 27). In all panels, * *p* < 0.05; *** *p* < 0.001, two-tailed *t*-test. Detailed information about the Western blots can be found in Appendix A.

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
