# Peer review of "Mdm2-Mediated Downmodulation of GRK2 Restricts Centrosome Separation for Proper Chromosome Congression"

_cells, 2021, doi:10.3390/cells10040729_

Round 1
Reviewer 1 Report
The authors study the regulation of centrosome disjunction, the result of intercentrosomal linker disassembly. This step of the centrosomal cycle facilitates the physical separation of the centrosomes and the correct formation of the mitotic spindle, and is tightly regulated by phosphorylation. Under the control of epidermal growth factor receptors, the G protein–coupled receptor kinase GRK2 has been shown to phosphorylates Mst2, in turn resulting in the activation of the NIMA kinase Nek2A, the major controller of intercentrosomal linker stability, through a convoluted mechanism that involves the phosphatase PP1 gamma.
The authors describe the importance of GRK2 for the temporal regulation of centrosomal disjunction. Their findings suggest that GRK2 is degraded in G2 through the action of Mdm2, an E3 ligase previously described by the authors to be responsible for the ubiquitination of GRK2 upon stimulation of G protein coupled receptors. This degradation is proposed to restrain centrosome disjunction and thus subsequent centrosome separation. Phosphorylation of GRK2 at a specific site, Ser670, is necessary for ubiquitination and degradation of the kinase. This phosphorylation additionally modulates GRK2 activity and specificity, suggesting that GRK2 is tightly regulated in order to properly control centrosome disjunction in G2.
The experiments are well done and clearly presented, with adequate quantifications and statistics when needed. I would agree that they may suggest that the cellular amount of GRK2 has a role controlling centrosome separation and that Ser670 is important for this. Indeed the authors show that interference with GRK2 degradation (through the modulation of Mdm2 activity and the expression of GRK2 Ser670 mutants) results in abnormal centrosome separation. But in my opinion it is difficult to draw conclusions from some of the experiments, and additional data would be needed to grant the acceptance of the paper. Specifically, a clearer proof that GRK2 is degraded and ubiquitinated specifically in G2 should be produced.
Major points:
- Etoposide is used to arrest cells in G2. As this drug obviously causes DNA damage, it would be desirable to see whether other forms of G2 arrest that do not affect DNA integrity (i.e. CDK1 inhibition by RO-3306) also induce GRK2 degradation (as indeed is suggested by synchronization experiments).
- Figure 1 (and other figures): besides the geminin and H3-P blots, DNA content as assessed by FACS should be shown to demonstrate that cells are indeed in the assigned phases of the cell cycle (ideally combined with H3-P detection to quantify the percentage of mitotic cells in the G2/M peak).
- GRK2 degradation seems to coincide with the peak of H3 phosphorylation, usually happening in mitosis, not in G2. Comment.
- Figure 1C should be part of 1B and show an example of the westren blots quantified. I fail to see them in the supplementary materials.
- Why is the western in Figure 2A cut? The data (Figure S2) seem to show almost no variation in the levels of GRK2 in parental cells.
- Figure 2B, the western blot does not seem to support the quantification.
- I fail to see ubiquitinated GRK2 in figure 2C. Several strong bands and a smear at high MW are observed in the ubiquitin western in the lane that lacks GRK2. A more clear experiment should be presented.
- Is proteasome inhibition recovering GRK2 amounts in G2? does it result in an accumulation of high molecular weight ubiquitinated GRK2 forms?
- Figure 3C, 4D, if geminin “specifically accumulates in G2” (page 5), why does the figure show higher levels of the protein 0.5 hours post-G1/S release?
- Besides FACS data an additional manner to show convincingly that cells are in G2 in Figure 3 (and 4) would be to co-stain cells with anti-cyclin B.
- Figure 4: I fail to see how Mdm2 inhibition could fail to interfere with centrosome separation in an asynchronously growing culture (where only cells in G2 would separate the centrosomes) but do so in cells synchronously going through G2.
- Figures 4B and 4C should use the same type of graphic.
- Figure 4 should show that the cells are indeed in G2 (or any other phase of the cell cycle, asynchronously growing in 4A, if I understood correctly). This is specially important here to discard the possibility that HLI373 affects cell cycle progression and distribution.
Minor:
- line 43: What is the “S-M linker”? Do the authors mean the Daughter-Mother linker?
- line 216: “Levels of Geminin, a protein that specifically accumulates in G2”, a reference should be provided.
- The observed degradation of GRK2 in G2 is not total (possibly ~25% diminution in total amounts). The authors could comment on how such a limited change in the amount of an enzyme that by definition acts catalytically could result in a physiological response (i.e. is GRK2 specifically degraded in G2 at the centrosome?).
- Figure 4C would benefit from arrows indicating where the centrosomes are.
Reviewer 2 Report
This study showed that GRK2 plays an essential role in centrosome separation. GRK2 phosphorylated by CDK2 phosphorylates Mst2 leading to Nek2A activation and centrosome separation. Phosphorylation by CDK2 also enhances Mdm2-mediated downregulation of GRK2. This study has interesting findings and contributes to understanding of centrosome regulation by GRK2. The comments are followed. 1. Fig. 2B shows that spindle length is regulated by GRK2. The results are significant. However, it is difficult to recognize the points used for evaluation. It is better to mark the points used for evaluation by arrow. Then, it is easily to understand what differences are. 2. S670A-GRK2 does not phosphorylate Mst2. Is this low kinase activity of GRK2, reduced affinity of GRK2 for Mst2, or other? The reason should be discussed in more detail. 3. Concerned with Mst2 phosphorylation by GRK2, authors describe ‘Mst2 is EGF-dependent biased substrate’ at line 418. What is biased substrate? This new terminology should be explained in more detail. 4. The mechanistic relationship between S670 phosphorylation of GRK2 and modulation of HDAC6 activity is better to include in discussion. So far, the effects of S670 phosphorylation of GRK2 on HADC6 function is not clearly described in the context of GRK2-mediated centrosome separation. 5. It is better to include the overall scheme of GRK2-mdiated centrosome regulation in supplement.Author Response
Please see the attachment

Round 2
Reviewer 1 Report
The authors have somehow addressed my concerns, for the most part adequately. Importantly, they now show data that suggests that GRK2 is ubiquitinated in G2, and that this modification depends on the phosphorylation of GRK2 Ser670.
The manuscript is now in my opinion suitable for publication in the journal.